# Role of Soluble ST2 Biomarker in Predicting Recurrence of Atrial Fibrillation after Electrical Cardioversion or Pulmonary Vein Isolation

**DOI:** 10.3390/ijms241814045

**Published:** 2023-09-13

**Authors:** Javier García-Seara, Laila González Melchor, Javier Rodríguez García, Francisco Gude, José Luis Martínez Sande, Moisés Rodríguez Mañero, Xesús Alberte Fernández López, Carlos Minguito Carazo, Teba González Ferrero, Sonia Eiras, Ricardo Lage, Isabel Moscoso, Sandra Feijoo Bandín, Francisca Lago, Ezequiel Alvarez, Clara V. Alvarez, José Ramón González Juanatey

**Affiliations:** 1Cardiology Department, University Clinical Hospital of Santiago de Compostela, 15706 Santiago de Compostela, Spain; laila.gonzalez.melchor@sergas.es (L.G.M.); jose.luis.martinez.sande@sergas.es (J.L.M.S.); moises.rodriguez.manero@sergas.es (M.R.M.); xesus.alberte.fernandez.lopez@sergas.es (X.A.F.L.); carlos.minguito.carazo@sergas.es (C.M.C.); teba.gonzalez.ferrero@sergas.es (T.G.F.); jose.ramon.gonzalez.juanatey@sergas.es (J.R.G.J.); 2Centro de Investigación Biomédica Cardiovascular en Red (CIBERCV), Institute of Health Carlos III, C/Monforte de Lemos 3-5, Pabellón 11, Planta 0, 28029 Madrid, Spain; sonia.eiras.pena@sergas.es (S.E.); ricardo.lage@sergas.es (R.L.); isabel.moscoso@sergas.es (I.M.); sandra.feijoo.bandin@sergas.es (S.F.B.); francisca.lago@sergas.es (F.L.); 3Institute of Biomedical Research of Santiago de Compostela (IDIS-SERGAS), Travesía da Choupana s/n, 15706 Santiago de Compostela, Spain; francisco.gude.sampedro@sergas.es; 4Clinical Analysis Department, University Clinical Hospital of Santiago de Compostela, 15706 Santiago de Compostela, Spain; javier.rodriguez.garcia@sergas.es; 5Epidemiology and Biostatistics Unit, University Clinical Hospital of Santiago de Compostela, 15706 Santiago de Compostela, Spain; 6Cardiology Translational Group, Health Research Institute of Santiago de Compostela, 15706 Santiago de Compostela, Spain; 7Cardiology Group, Centre for Research in Molecular Medicine and Chronic Diseases (CIMUS), Universidade de Santiago de Compostela, 15782 Santiago de Compostela, Spain; 8Cellular and Molecular Cardiology Unit and Department of Cardiology, Institute of Biomedical Research of Santiago de Compostela (IDIS-SERGAS), Travesía da Choupana s/n, 15706 Santiago de Compostela, Spain; 9Department of Pharmacology, Pharmacy and Pharmaceutical Technology, University of Santiago de Compostela, 15782 Santiago de Compostela, Spain; ezequiel.alvarez@sergas.es; 10Neoplasia & Endocrine Differentiation, Centro de Investigación en Medicina Molecular y Enfermedades Crónicas (CIMUS), University of Santiago de Compostela, Instituto de Investigación Sanitaria (IDIS), 15782 Santiago de Compostela, Spain; claravalvarez@usc.es

**Keywords:** soluble suppression of tumorigenicity 2 (sST2), atrial fibrillation, electrical cardioversion, pulmonary vein ablation

## Abstract

This study aims to determine the predictive value of the soluble suppression of tumorigenicity 2 (sST2) biomarker in atrial fibrillation (AF) recurrence. This prospective, observational study included patients with AF referred for electrical cardioversion (ECV) or pulmonary vein isolation (PVI) procedures. Baseline characteristics were collected, and sST2 was determined at baseline and at 3 and 6 months of follow-up. sST2 was determined at baseline in a matched control group. Left atrial voltage mapping was performed in patients undergoing PVI. The sST2 maximal predictive capacity of AF recurrence was at the 3-month FU in the cohort of patients undergoing ECV with respect to 6-month AF recurrence with an AUC of 0.669, a cut-off point of 15,511 pg/mL, a sensitivity of 60.97%, and a specificity of 69.81%. The ROC curve of the sST2 biomarker at baseline and 3 months in the cohort of patients undergoing PVI showed AUCs of 0.539 and 0.490, respectively. The logistic regression model identified the rhythm (AF) and the sST2 biomarker at 3 months as independent factors for recurrence at 6 months in the ECV cohort. In the logistic regression model, sST2 was not an independent factor for recurrence at 6 months of follow-up in the PVI cohort. In patients who underwent ECV, sST2 values at 3 months may provide utility to predict AF recurrence at 6 months of follow-up. In patients who underwent PVI, sST2 had no value in predicting AF recurrence at 6 months of follow-up.

## 1. Introduction

The suppression of tumorigenicity 2 (ST2) protein is encoded by the interleukin-1 receptor-like 1 (IL1RL1) gene [1]. ST2 belongs to the interleukin-1 receptor family, comprising a transmembrane receptor form (LST2) and a truncated soluble receptor form (sST2), detectable in serum. sST2 functions as a decoy by binding to IL-33, thereby inhibiting the IL-33/ST2 signaling pathway in a dose-dependent manner, a process linked to fibrosis. This phenomenon has been extensively elucidated within the context of inflammatory and immune disorders [2]. Notably, in 2002, the association of LST2 and sST2 with cardiomyocytes and fibroblasts was established [3]. Under mechanical strain, ST2 experiences significant upregulation and contributes to the myocardial response to biomechanical overload in cardiomyocytes during heart failure. The inducible expression of sST2 is ubiquitous, observed across nearly all cell types [4,5].

The application of sST2 as a biomarker has found utility in inflammatory diseases and, more recently, in heart failure due to its sensitivity to factors such as inflammation, stretching, and fibrosis [6,7,8]. Given the distinct attributes of this biomarker, we posit its clinical significance as a prognostic indicator for AF recurrence.

Previous studies investigating the predictive value of sST2 in AF recurrence have predominantly relied on subgroup analyses of patients with heart failure. Notably, studies specifically tailored to assess sST2’s prognostic efficacy in AF have primarily focused on baseline measurements, with limited exploration into the dynamics over time [9,10].

In this investigation, our primary objective was to evaluate the clinical utility of the sST2 biomarker among patients afflicted with AF. To achieve this, we adopted a comprehensive approach by obtaining measurements at baseline, 3 months, and 6 months of follow-up. This extended evaluation aimed to enhance the precision of the sST2 biomarker’s clinical efficacy in predicting AF recurrence. Our study cohort encompassed individuals undergoing either electrical cardioversion (ECV) or pulmonary vein ablation (PVI).

## 2. Results

### 2.1. Baseline Characteristics

A total of 307 patients diagnosed with AF between 1 September 2016 and 30 September 2019 were included. Thirty-seven patients were excluded because they did not meet the inclusion and/or exclusion criteria as follows: five with mitral valve prosthesis, three with aortic valve prosthesis, seven with moderate–severe left ventricular dysfunction, four with hypertrophic cardiomyopathy, one with calcific pericarditis, three with rheumatoid arthritis, one with pulmonary sarcoidosis, one with renal transplantation, two with severe bronchial asthma, one with severe mitral regurgitation, one with multiple myeloma under treatment, two with pacemakers, two with ulcerative colitis, three with skin psoriasis, and one with psoriatic arthritis.

Of the 270 patients included, 20 patients did not undergo complete follow-up (at 3 and/or 6 months), and 250 patients were finally analyzed. Ninety-four patients underwent ECV, and one hundred and fifty-six patients underwent electrophysiological study with PVI. In addition, 55 control patients were included (Figure 1).

### 2.2. Characteristics of Patients with Atrial Fibrillation vs. Those of Control Participants

AF patients were compared with controls matched for age and sex. Patients with AF had higher BMI, hypertension, tachycardiomyopathy, and right and left atrial dimensions and volume without differences in LVEF (Table 1). The blood glucose and creatinine values were also significantly higher in AF patients. At baseline, the majority of patients were on antiarrhythmic drugs (92 patients, 97.8% in the ECV group and 152 patients, 97.4% in the PVI group). At the 3-month follow-up, 85 patients, 90.4% in the ECV group and 128 patients, 82.0% in the PVI group, remained on antiarrhythmic drugs. The baseline biomarker range (sST2.0) in controls was slightly lower in women, [2220–17,286 pg/mL], vs. [3895–29,961 pg/mL] in men. A normal distribution of the sST2.0 biomarker was demonstrated in controls and in patients with AF. The mean sST2.0 values were significantly higher [14,677 ± 8214 pg/mL] in the total AF group than in the control group [11,016 ± 5618 pg/mL], *p* = 0.007.

### 2.3. Characteristics of Patients According to the Procedure Performed

The patients who underwent ECV presented significantly higher values of sST2.0 (17,163 ± 914 pg/mL) than the control participants (11,016 ± 5547 pg/mL), *p* = 0.001. However, patients who underwent PVI presented baseline values of sST2 similar to those of controls (13,178 ± 7223 vs. 11,016 ± 5547), *p* = 0.07. Patients who underwent ECV had significantly higher baseline values of sST2.0 than patients who underwent PVI (17,163 ± 914 pg/mL vs. 13,178 ± 7223), *p* = 0.000 (Table 1). Patients in sinus rhythm at baseline also had significantly higher baseline sST2 values than those in AF. Patients with persistent AF had significantly higher baseline sST2 values than those with paroxysmal AF (15,634 ± 8651 vs. 12,113 ± 6278; *p* = 0.002) (Table 1).

Patients with a ventricular rate > 100 bpm tended to have higher sST2.0 values than those with a ventricular rate < 100 bpm. These differences were significant in the persistent AF group (Appendix A).

### 2.4. Recurrence of AF and ST2S Biomarker

The sST2 biomarker results were analyzed according to the presence or absence of recurrence at follow-up (from basal to 3 months of follow-up, 3 to 6 months of follow-up, and up to 6 months of follow-up), as shown in Table 2.

### 2.5. Recurrence of AF and sST2 Biomarker Patients Undergoing CVE

The sST2 biomarker values were significantly lower at the 3-month and 6-month follow-ups when no AF recurrence was observed. The greatest reduction occurred in the first 3 months, with no additional significant reductions in the biomarker observed between 3 and 6 months in the absence of AF recurrence. (Table 3 and Figure 2).

### 2.6. Recurrence of AF and sST2 in Patients Undergoing PVI

The values of the sST2 biomarker were slightly higher at the 3-month follow-up in the group with no AF recurrence, without these differences reaching statistical significance. At 6 months, the sST2 values were significantly higher in patients who did not present with AF recurrence (Table 4 and Figure 3). These differences were analyzed according to the type of AF, and a greater elevation in the paroxysmal form was observed.

No significant differences were found in the value of the biomarker in AF recurrence depending on the type of persistent or paroxysmal AF. The biomarker elevation was greater in the paroxysmal form in the first 3 months (Appendix A).

### 2.7. Recurrence of AF and sST2 Biomarker in the Global Cohort Excluding Patients Who Underwent a Procedure during Follow-Up

Differences in the sST2 biomarker in recurrence at follow-up in patients with AF were analyzed, excluding patients who had undergone an ECV or PVI procedure during follow-up. Thirty patients were excluded from this analysis.

Patients without AF recurrence at 3 months had a significantly greater decrease than those with AF recurrence in this period. The patients who did not recur in the period of 3 to 6 months had a greater but not statistically significant decrease in sST2 than those who did in this period of time with respect to the baseline sST2 (Appendix A).

### 2.8. Predictive Capacity of the sST2 Biomarker for AF Recurrence

To determine the discriminative capacity of the sST2 biomarker with respect to recurrence, a receiver operating characteristic curve (ROC) analysis was performed and the area under the curve (AUC) of the biomarker was obtained.

#### 2.8.1. ROC Curve of the Baseline Biomarker (sST2.0) in the Cohort of Patients Undergoing ECV and Recurrence at 3 Months

The ROC curve of the baseline sST2 biomarker of the cohort of patients undergoing ECV with respect to recurrence at 3 months was calculated, and the AUC was 0.419 (Figure 4A).

#### 2.8.2. ROC Curve of the Biomarker sST2 at 3-Month Follow-Up (sST2.1) in the Cohort of Patients Undergoing ECV and Recurrence from 3 to 6 Months

The ROC curve of the sST2 biomarker was calculated at the 3-month follow-up in the cohort of patients undergoing ECV with respect to 6-month recurrence, and the AUC was 0.669 (Figure 4B).

The value of the sST2 biomarker at 3 months showed the best predictive capacity for AF recurrence at 6 months in cases undergoing ECV. The cut-off point of the sST2 biomarker at 3 months as a predictor of AF recurrence at 6 months in the cohort of patients undergoing ECV was 15,511 pg/mL with a sensitivity of 60.97% and a specificity of 69.81%.

#### 2.8.3. ROC Curve of the Biomarker sST2.0 in the Cohort of Patients Undergoing PVI and Recurrence at 3 Months

The ROC curve of the baseline sST2 biomarker of PVI with respect to recurrence at 3 months was calculated, and the AUC was 0.539 (Figure 5A).

#### 2.8.4. ROC Curve of the Biomarker ST2S.1 in the Cohort of Patients Undergoing PVI and Recurrence from 3 to 6 Months

The ROC curve of the sST2 biomarker at 3 months was calculated in the cohort of patients undergoing PVI with respect to recurrence at 6 months, and the AUC was 0.490 (Figure 5B).

### 2.9. Logistic Regression Models to Predict AF Recurrence

#### 2.9.1. Logistic Regression Models in Patients Undergoing ECV


*AF recurrence at 3 months:*


In the cohort of patients undergoing ECV, the logistic regression models of the variables analyzed only identified the indexed volume of LA as an independent factor of recurrence at 3 months (Table 5(A)).


*AF recurrence at 6 months:*


In the cohort of patients undergoing ECV, the Cox regression model of the variables analyzed identified the rhythm (AF at 3 months) and the sST2 biomarker at the first follow-up (3 months) as independent factors for recurrence at 6 months (Table 5(B)).

#### 2.9.2. Logistic Regression Models in Patients Undergoing PVI


*AF recurrence at 3 months:*


In the cohort of patients undergoing PVI, the Cox regression model of the variables analyzed identified heart rate, rhythm (AF), and LA indexed volume as independent factors for recurrence (Table 6(A)).


*AF recurrence at 6 months:*


In the cohort of patients undergoing PVI, the Cox regression model of the variables analyzed identified only the rhythm variable (AF at 3 months) as an independent factor for recurrence (Table 6(B)).

### 2.10. sST2 Biomarker and LA Low-Voltage Areas

To analyze the value of the sST2 biomarker with the low-voltage area in LA, the results were classified into three groups. As shown in Table 5, patients with a low-voltage area of less than 5% and >35% had lower sST2.0 values than patients with an intermediate low-voltage area percentage, although these differences were not statistically significant (*p* = 0.293) (Table 7).

## 3. Discussion

These are the main findings of our study:

### 3.1. Baseline sST2 in AF Patients and Controls

The baseline sST2 biomarker is higher in patients with AF than in controls, although these differences are concentrated in patients with the persistent form of AF. This finding is consistent with the result published in a study of 92 patients with persistent AF [11]. However, patients with paroxysmal AF did not present significant differences in the biomarker sST2 compared to controls. This result contrasts with the data from another study in which the biomarker was analyzed in a series of 174 patients with AF to determine its ability to predict the development of heart failure at 6 months of follow-up and where higher levels of the sST2 biomarker were described in paroxysmal AF with respect to control cases [12]. In that study, the patients were recruited from hospitalization, and 26.5% developed heart failure during follow-up, unlike our series in which all recruited patients were ambulatory and 12% had previously presented tachycardiomyopathy, but all had recovered normal systolic LV function. The baseline sST2 biomarker rose significantly more in patients with AF than in patients with sinus rhythm at baseline, as well as in those with >100 bpm with persistent AF. Both characteristics suggest that the hemodynamics (high heart rate and overload due to atrial myopathy) linked to the time in AF (persistent) are the predominant factors in the basal sST2 levels.

### 3.2. Baseline sST2 and AF Recurrence

Baseline levels of the sST2 biomarker did not have a predictive capacity for AF recurrence in either the cohort of patients undergoing ECV or the cohort of patients undergoing PVI. This finding is consistent with the results of a study of 115 patients who underwent ECV, in which the baseline value of the sST2 biomarker was not a predictor of recurrence [13]. It is noteworthy that the substantial baseline variability of the sST2 biomarker does not correlate with subsequent AF recurrence, which makes it very difficult to be used as a predictor of recurrence in patients with ECV (Figure 2). A substantial baseline variability of the sST2 biomarker is also observed in patients undergoing PVI; thus, it is difficult to establish a baseline level of the biomarker that correlates with recurrence. This finding contrasts with the result of a publication of a series of 100 patients who underwent cryoablation and another of 258 patients who underwent radiofrequency ablation, in which the baseline value of sST2 was shown to be a predictor of recurrence [10,14]. The most notable differences with respect to our series are that the follow-up was longer (1 year) and the provider of sST2 was different. In our series, the baseline sST2 values were very heterogeneous, and although there was a tendency for lower values to have less recurrence, the baseline sST2 did not show predictive capacity for AF recurrence.

### 3.3. sST2 at 3 Months of Follow-Up and AF Recurrence

The levels of the sST2 biomarker at 3 months of follow-up were predictive of AF recurrence at 6 months in patients undergoing ECV, with a statistical value c: 0.669. This finding was clinically relevant in our study. Patients undergoing ECV who did not experience recurrence had a significantly reduced sST2 value at 3 months, such that the reduction after 6 months, if non-recurrence continued, was very small. Three months after ECV, the baseline heterogeneity of sST2 disappeared, probably caused by baseline hemodynamic factors, and the sST2 value predicted recurrence with a cut-off of 15,511 pg/mL with a sensitivity of 60.97% and a specificity of 69.81%.

In contrast, in patients undergoing PVI, the levels of the sST2 biomarker at 3 months presented a generalized elevation and did not have an adequate predictive capacity for AF recurrence. The sST2 biomarker rose during the first 3 months, regardless of AF recurrence, in patients undergoing PVI, confirming the inflammatory stage of the blanking period. This rise was more pronounced in patients with paroxysmal AF who started from lower baseline figures than in patients with persistent AF. Baseline sST2 values in the cohort of patients undergoing PVI were not predictors of AF recurrence. This result differs from what was previously published in a study of 100 patients who underwent cryoablation for paroxysmal AF. A baseline value of sST2 was a predictor of AF recurrence at 1 year with a cut-off point of 30,600 pg/mL and an area under the curve of 0.831 [14]. In our series, the absence of differences between patients with paroxysmal AF and controls stands out, and in patients undergoing PVI, when analyzing the levels of the sST2 biomarker at the first and second follow-up, they increased, especially in those cases without recurrence, and no significant differences were observed in those who recurred. Similarly, when analyzing patients with paroxysmal AF, the values of the sST2 biomarker (even when excluding paroxysmal recurrences and patients who underwent follow-up procedures) showed a significant elevation at the first and second follow-up with respect to the baseline value in those who did not recur, and the values did not change significantly in the cases with recurrence. These results are different from a previously published study of 23 patients in which higher baseline values of the sST2 biomarker combined with higher values of troponin T at an approximate 1-year follow-up were predictors of recurrence after PVI [10]. Perhaps the long-term follow-up can explain the differences in the results with respect to our cohort, where the maximum follow-up was 6 months. Additionally, the differences in the technique used (cryoablation vs. radiofrequency) could play a role in the differences. However, in another series of 258 patients who underwent radiofrequency ablation of AF, the baseline sST2 values were significantly higher (31.2 ng/mL) in those who experienced recurrence after 1 year than in those who did not experience recurrence (20.3 ng/mL). It should be noted that in the patients in whom a second ablation was performed and new alterations were detected using endocardial mapping, the baseline sST2 was higher (43.0 ng/mL vs. 22.1 ng/mL in the AF recurrence subgroup without new alterations in endocardial mapping); therefore, the biomarker could be useful in predicting a more unfavorable structural evolution than AF recurrence [15]. The baseline sST2 values in our series were very heterogeneous, and although there was a trend in which patients with lower values had fewer recurrences, they did not seem to have a predictive value.

### 3.4. sST2 in PVI Patients and Fibrosis

The levels of the sST2 biomarker did not have a statistically significant association with respect to the low-voltage area in the electroanatomical voltage mapping of the LA in patients undergoing PVI. The low-voltage areas determined using the maps obtained with high-point-density catheters were used as surrogate markers of interstitial fibrosis [16]. Higher values of the biomarker were observed in patients with intermediate levels of low-voltage areas and lower levels in those with normal voltage or very high percentages of low-voltage areas, although none of these results were significantly different. Previous studies of the sST2 biomarker in other heart diseases, such as the subanalysis of the PARADIGM-HF study [17,18], demonstrated a relationship between this biomarker and ventricular remodeling in patients with heart failure under treatment with sacubitril/valsartan in an 8-month follow-up. A relationship has been also described between replacement myocardial fibrosis and sST2 circulation levels in patients with severe aortic stenosis [19]. However, there is no evidence of this correlation between sST2 and atrial fibrosis. It is noteworthy that patients with a higher percentage of low-voltage areas had lower baseline sST2 values than patients with an intermediate percentage of low-voltage areas, suggesting that inflammation and distension were better factors than atrial fibrosis for determining blood levels of sST2. It seems that the type of fibrosis and its location within the atrial wall have a great impact in the resulting arrhythmogenic effect on atrial conduction. In this way, endomysial fibrosis tissue (fibrosis within individual myocites) rather than fibrosis tissue surrounding bundles of myocites is responsible for the increased complexity of fibrillation conduction. Then, the overall fibrosis tissue content would not be the main determinant of conduction disturbance in AF [20]. It seems that the hemodynamic factor most influences the basal level of sST2, and the inflammatory factor predominates in the first months in the levels of sST2 in patients with AF undergoing PVI. There is increasing evidence that there is a link between oxidative processes and AF and various inflammatory markers have been associated with AF. These mediators can reflect changes in atrial electrophysiology and structural substrates, which lead to increased vulnerability to AF [21].

A probable explanation for our results is that sST2, in the same way as other biomarkers associated with inflammation, could be elevated in the first 3 to 6 months after PVI in the blanking period, which lasts approximately 3 months after ablation [22,23]. Similarly, in patients with paroxysmal AF who had a lower baseline biomarker value but a higher elevation at the first follow-up in cases without AF recurrence, it could be due to a greater inflammatory response than in patients with persistent AF who present greater atrial remodeling [24,25] or in patients with recurrence of AF.

### 3.5. Limitations

This study was observational and unicentric with a relatively small sample size. A larger population is necessary to confirm the results. Serum marker sST2 is not heart-specific and the findings were not obtained from coronary sinus sampling. We included one patient with heart failure with preserved LVEF in our study. The treatment for heart failure could potentially have influenced sST2 values independently of atrial fibrillation recurrence.

## 4. Methods and Materials

### 4.1. Study Design

This was a prospective, descriptive, observational unicentric study that included all patients with AF referred to the Arrhythmia Unit of the Cardiology Department of University Hospital of Santiago de Compostela who underwent an ECV or PVI procedure from 1 September 2016 to 30 September 2019. All patients underwent a laboratory analysis, ECG, and transthoracic echocardiogram with a 2–4 MHz transducer (Siemens Medical solutions, Erlangen, Germany) with the following echocardiographic variables measured: left atrium (LA) anteroposterior diameter, LA area, LA indexed volume, right atrium (RA) area, RA index volume, left ventricular ejection fraction (LVEF), LV end-systolic diameter, LV end-diastolic diameter, LV end-systolic volume, LV end-diastolic volume, right ventricular (RV) end-diastolic diameter, and RV end-systolic diameter.

We performed measurements of sST2 at baseline, 3 months, and 6 months of follow-up in AF patients and only at baseline in controls.

The control group consisted of patients referred to the arrhythmia unit for an electrophysiological study other than AF ECV or PVI during the same period of time. The control patients did not have AF.

### 4.2. Study Population

#### 4.2.1. Inclusion Criteria

We included all patients referred to the arrhythmia unit during the time period indicated above for an ECV or PVI due to AF. All the patients signed an informed consent prior to being included in the study and having their data collected.

#### 4.2.2. Exclusion Criteria

All patients with any condition that could alter the sST2 biomarker values were excluded. Patients with any type of autoimmune or inflammatory chronic disease and those diagnosed with any structural heart disease or heart failure with systolic dysfunction were excluded with the aim of exploring the sST2 biomarker without the interaction of concomitant heart failure. Patients suffering from previous tachycardiomyopathy were excluded only if moderate or severe LV dysfunction was present at the beginning of the study. Systolic dysfunction was defined as LVEF < 50% using the Simpson method. Patients with moderate to severe chronic kidney disease, moderate to severe pulmonary disease, moderate to severe pulmonary hypertension, and thyroid disorders were excluded.

### 4.3. sST2 Measurement

The Quantikine Human sST2/IL33R Inmunoassay is a 4.5 h solid-phase ELISA designed to measure human sST2 in cell culture supernatants, serum, and plasma. We took peripheral blood samples obtained the same day of the procedure, at the 3-month follow-up visit, and at the 6-month follow-up visit. The blood extraction was at least 1 mL, and we used tubes without heparin to allow clotting for 30 min at room temperature before centrifugation for 15 min at 1000× *g*. We removed the serum and aliquoted and stored the samples at −80 °C. The aliquots were anonymized and codified such that all the analyses were performed in a blinded fashion.

For the sST2 analysis, the aliquots were thawed and required a 20-fold dilution. We ultimately determined the optical density of each well within 30 min using a microplate reader set to 450 nm.

### 4.4. AF Ablation/Electrical Cardioversion

The ablation procedure was performed under conscious sedation. Transeptal punction was conducted via a femoral approach. A decapolar catheter 6F (Dynamic Deca Bard Electrophysiology, Lowell, MA, USA) was positioned in the coronary sinus and a quadripolar catheter was positioned in His (Biosense Webster Inc., Irvine CA, USA). Transeptal puncture was guided with fluoroscopy and anatomical references using a SL0 sheath (63 cm, Swartz Abbott, Plymouth, MN, USA) and a transeptal needle (BRK XS Series, 71 cm, Abbot St Jude Medical, Minnetonka, MN, USA). The procedure was performed under the anticoagulation effect of intravenous heparin. After transeptal puncture, the amount of heparin was adjusted to achieve an activated clotting time between 300 s and 350 s.

A three-dimensional electroanatomical reconstruction of the left atrium and pulmonary veins was created with the CARTO3 system (Biosense Webster Inc., Irvine, CA, USA). Bipolar voltage mapping was performed in sinus rhythm with a high-density catheter (Biosense Webster Pentaray Inc., Irvine, CA, USA). The pulmonary vein and left atrial appendage areas were ruled out. We used the following classification for the voltage mapping: >1.5 mV, normal voltage; 0.5–1.5 mV, intermediate range; <0.5 mV, scar. Based on the Utah classification, the low-voltage mapping area was classified into 4 grades: grade 1 (<5% scar), grade 2 (5–20% scar), grade 3 (20–35% scar), and grade 4 (≥35% scar).

Electrical cardioversion was performed under propofol (1 mg/kg dose) with the patient properly anticoagulated during the three prior weeks. If the patient was not properly anticoagulated, we performed a transesophageal echocardiogram to exclude left atrial thrombus. We delivered a synchronized 360-joule biphasic shock to restore the sinus rhythm.

### 4.5. Follow-Up

Anti-arrhythmic drugs were maintained for at least 3 months after ablation and indefinitely or at physician discretion after ECV. Lifetime anticoagulation would be recommended in patients with CHA_2_DS_2_-VASc scores ≥ 2. Otherwise, anticoagulation was stopped after 2 months in patients undergoing PVI and after 1 month in patients undergoing ECV. All patients underwent ECG and 24 h Holter monitoring at 3 and 6 months. sST2 measurements were performed at baseline and at the 3- and 6-month follow-ups.

### 4.6. Statistical Analysis

Continuous variables are described as the mean ± SD if normally distributed, and a *t* test was used for comparisons between groups. For biomarker normality distribution analysis, the values obtained were transformed into logarithmic values. Categorical variables are expressed as frequencies and percentages, and the *χ*^2^ test or Fisher’s precision test was applied for comparisons between groups.

The results were expressed by a *p* value with 95% confidence intervals and odds ratio (OR). Statistical significance was defined as a *p* value < 0.05. Single-factor and multivariate logistic regression analyses were used to analyze the factors related to AF recurrence.

In the multivariate analysis for the construction of the models, we used generalized additive regression models (generalized additive models, GAMs) [26,27]. To estimate the effects of the continuous covariates, penalized low-rank thin-plate splines were used, using the criterion of restricted maximum likelihood (REML) for automatic selection of the degree of smoothing. The inference of the GAMs was carried out using the P-IRLS (Penalized Iteratively Reweighted Least Squares) algorithm available in the R mgcv package 1.9-0 [28].

A receiver operating characteristic curve analysis was used to evaluate the diagnostic accuracy. The calculation of the cut-off point was carried out with Younde’s formula.

To detect the difference in the sST2 biomarker with a power of 80%, we used the mean values of the population of control cases at 10,000 pg/mL and the mean of the population in the AF cases at 12,000 pg/mL, with a standard deviation of 6000. Considering a *p* value of 0.05, the calculation of the sample size in the analysis of two independent samples was at least 91 cases in each arm [26].

### 4.7. Ethical Considerations

This study was carried out with the approval of the Galician Clinical Research Ethics Committee under number 2016/358.

The development of the project was carried out in accordance with the Declaration of Helsinki of the World Medical Association of 1964 and ratifications of the following assemblies (Tokyo, Venice, Hong Kong, and South Africa) on ethical principles for medical research on human beings.

The ownership of this study takes into account the available knowledge, as well as the applicable legal requirements, particularly Law 14/2007 on biomedical research and the Royal Decree 1716/2011 of November 18, in which the basic requirements for the authorization and operation of biobanks for the purposes of biomedical research, the treatment of biological samples of human origin, and the operation and organization of the National Registry of Biobanks for biomedical research is regulated. Additionally, ORDESAS/3470/2009 of December 16, which publishes the Guidelines on Observational Type Postauthorization studies for medicines for human use, and RD1090/2015 of December 4, which regulates clinical trials of medicines, the Ethics Committees for Research with Medicines, and the Spanish Registry of Clinical Studies were adhered to.

## 5. Conclusions

Baseline sST2 is not useful to predict AF recurrence at 3 months of follow-up in patients with AF who have undergone ECV or PVI. In patients who underwent ECV, the sST2 values and AF rhythm at 3 months of follow-up may provide utility to predict AF recurrence at 6 months of follow-up, and in patients who underwent PVI, sST2 at 3 months of follow-up had no value in predicting AF recurrence at 6 months of follow-up.

## Figures and Tables

**Figure 1 ijms-24-14045-f001:**
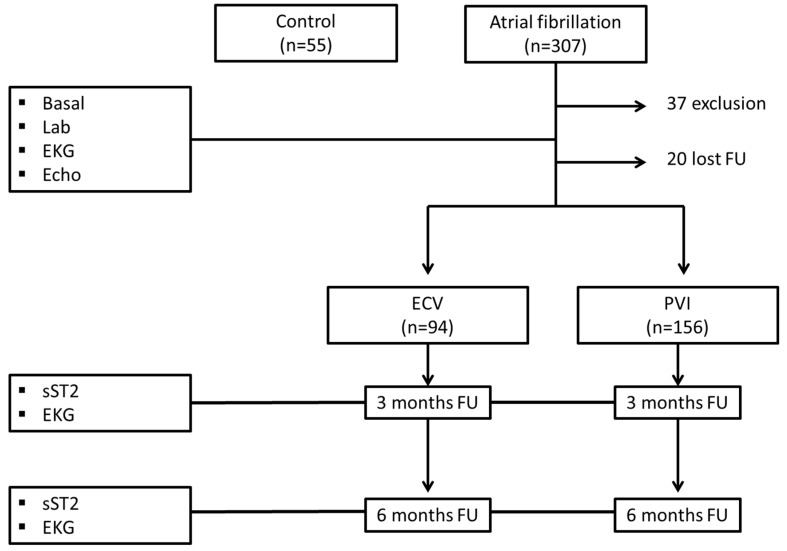
Flowchart of the study population. Lab.: laboratory test; EKG: electrocardiogram; Echo: echocardiogram; ECV: electrical cardioversion; PVI: pulmonary vein isolations; sST2: soluble ST2.

**Figure 2 ijms-24-14045-f002:**
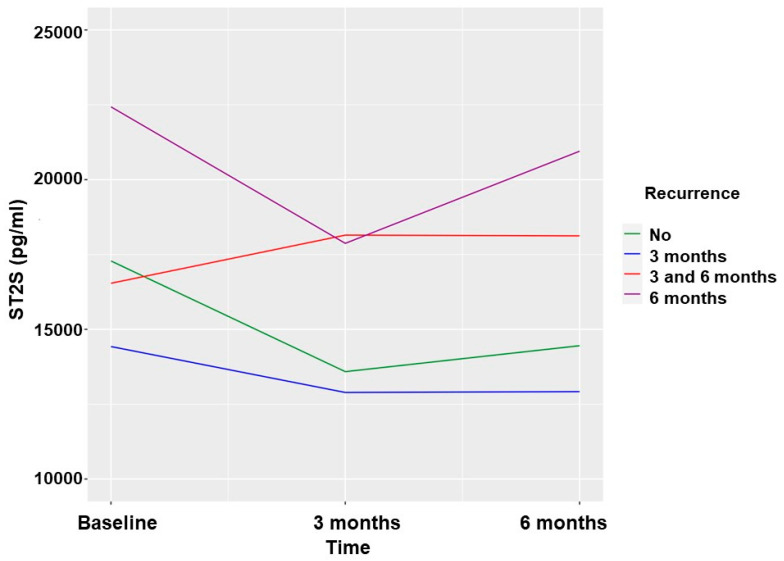
Soluble ST2 values according to AF recurrences in patients undergoing ECV.

**Figure 3 ijms-24-14045-f003:**
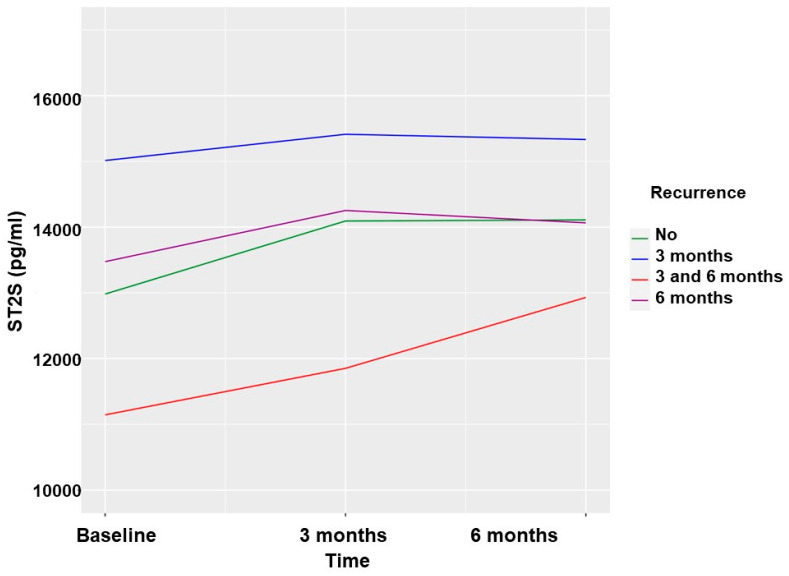
Soluble ST2 values according to AF recurrences in patients undergoing PVI.

**Figure 4 ijms-24-14045-f004:**
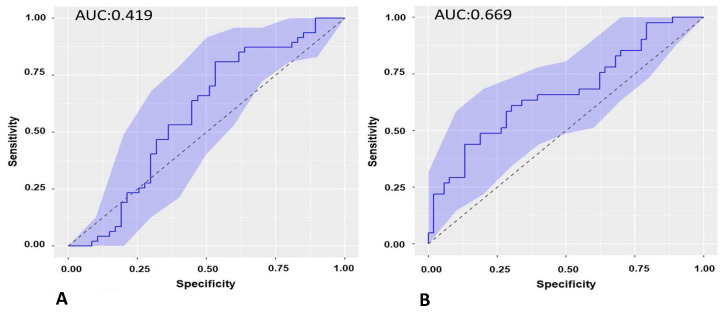
(**A**): ROC curve of the baseline sST2 in recurrence of AF at 3-month follow-up in the cohort of patients undergoing ECV. (**B**). ROC curve of sST2 at 3-month follow-up in recurrence of AF at 6-month follow-up in the cohort of patients undergoing ECV. ROC: receiver operating characteristic. ECV: electrical cardioversion.

**Figure 5 ijms-24-14045-f005:**
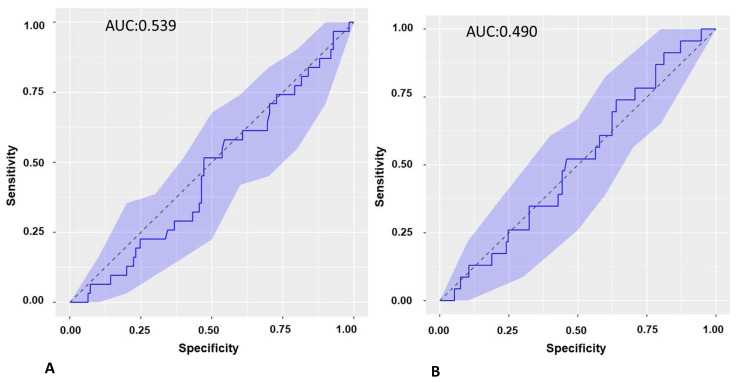
(**A**): ROC curve of baseline sST2 in recurrence of AF at 3-month follow-up in the cohort of patients undergoing PVI. (**B**): ROC curve of ST2S at 3-month follow-up in the recurrence of AF at 6-month follow-up in the cohort of patients undergoing PVI. ROC: receiver operating characteristic. PVI: pulmonary vein isolation.

**Table 1 ijms-24-14045-t001:** Baseline characteristics of the study population.

	ECV (n = 94)	PVI (n = 156)	P1	Control (n = 40)	P2
Age (years)	61.1 ± 9.1	56.9 ± 10.7	0.001 *	55.8 ± 11.3	0.068
Men (n, %)	74 (78.7)	110 (70.5)	0.154	25 (62.5)	0.146
Weight (kg)	86.5 ± 13.1	86.5 ± 14.7	0.494	74.6 ± 12.1	<0.000 *
Height	1.69 ± 0.09	1.69 ± 0.09	0.378	1.67 ± 0.09	0.094
BMI (kg/m^2^)	30.2 ± 4.3	30.3 ± 4.4	0.423	26.6 ± 2.89	<0.000 *
HT (n, %)	50 (53.2)	68 (43.6)	0.141	3 (7.5)	<0.000 *
DM (n, %)	10 (10.6)	13 (8.3)	0.541	2 (5)	0.379
Smoking (n, %)	22 (23.4)	47 (30.1)	0.249	11 (27.5)	0.989
COPD (n, %)	6 (6.4)	8 (5.1)	0.676	2 (5)	0.877
OSA (n, %)	5 (5.3)	9 (5.8)	0.881	0	0.230
CKD (n, %)	1 (1.1)	5 (3.2)	0.284	0	1
Obesity (n, %)	36 (38.3)	76 (48.7)	0.108	7 (17.5)	0.001 *
Tachycardiomyopathy (n, %)	12 (12.8)	19 (12.2)	0.892	0	0.011 *
Time from first AF diagnosis (months)	24.1 (32.2)	58.2 (70.1)	<0.000 *		
AF pattern (n, %) -Paroxysmal (n = 68) -Persistent (n = 182)					
	0 (0)	68 (100)	<0.000		
	94 (51.6)	88 (48.4)	0.234		
RA Area (cm^2^)	15.4 ± 4.2	13.6 ± 3.9	0.000 *	12.8 ± 3.1	0.001 *
LA AP Diameter (mm)	45.2 ± 5.5	41.2 ± 6.2	0.078383	23 ± 3.9	<0.000 *
LA Area (cm^2^)	24.2 ± 5	21.5 ± 5.4	0.090	13 ± 3.2	<0.000 *
LA Volume (mL)	96.2 ± 47.5	94.4 ± 54.3	0.397	53 ± 30.9	<0.000 *
LA Index Volume (mL/m^2^)	48.3 ± 22.5	47.9 ± 27.8	0.446	28.7 ± 16.8	<0.000 *
LVEF (%)	61.4 ± 7.2	62.6 ± 7.2	0.103	63 ± 6	0.176
LVEDV (mL)	60.4 ± 23.2	56.4 ± 25.6	0.106	59 ± 21.8	0.382
LVESV (mL)	23.2 ± 10.4	21.3 ± 10.8	0.092	22 ± 8.5	0.408
LVEDd (mm)	39.3 ± 7.1	38.9 ± 8.7	0.374	38 ± 7.1	0.349
LVESd (mm)	27.2 ± 4.3	26.9 ± 6.1	0.338	27 ± 4.8	0.415
RVEDd (mm)	26.7 ± 4.8	26.7 ± 4.7	0.496	25 ± 4.5	0.027 *
Glucose (mg/dL)	105.1 ± 23.3			98.3 ± 18	0.041 *
Urea (mg/dL)	45.6 ± 14.5			39.2 ± 10.8	0.005 *
Creatinine (mg/dL)	0.98 ± 0.30			0.80 ± 0.16	0.001 *
Hb1Ac (%)	6.9 ± 1.2			5.3 ± 0.7	0.001 *
TSH (mU/L)	2.9 ± 2.4			3.1 ± 5.7	0.232
AAD (n, %) -Amiodarone -Flecainide -Propafenone -Betablockers -Calcium antagonists	92 (97.8)	152 (97.4)	0.542	11(27.5)	0.001 *
	44 (46.8)	48 (30.8)			
	46 (48.9)	101 (64.7)			
	2 (2.1)	3 (1.9)			
	88 (93.6)	138 (88.5)			
	2 (2.1)	8 (5.1)			
sST2.0 (pg/mL)	17,163 ± 9147	13,178 ± 7223	0.000 *	11,016 ± 5618	0.007 *
sST2.0 and AF type (pg/mL) -Paroxysmal -Persistent -Persistent vs. paroxysmal					
	12,113 ± 6278			11,016 ± 5618	0.234
	15,634 ± 8651			11,016 ± 5618	0.011 *
	15,634 ± 8651 vs. 12,113 ± 6278		0.002 *		
sST2.0 and initial rhythm (pg/mL) -Sinus rhythm, n = 89 -AF, n = 161			0.003 *		
	12,626 ± 6483				
	15,810 ± 8846				
sST2.0 and rate (pg/mL) <100 bpm, n = 221 ≥100 bpm, n = 29			0.065		
	14,330 ± 7641				
	17,321 ± 11,534				
sST2.0 and rate in persistent AF (pg/mL) <100 bpm, n = 159 ≥100 bpm, n = 23			0.032 *		
	15,113 ± 7941				
	19,237 ± 12,136				

ECV: electrical cardioversion; PVI: pulmonary vein isolation; BMI: body mass index; HT: hypertension; DM: diabetes mellitus; COPD: chronic obstructive pulmonary disease; OSA: obstructive sleep apnea; CKD: chronic kidney disease; RA: right atrium; LA: left atrium; AP: anteroposterior; LVEF: left ventricular ejection fraction; LVEDD: left ventricular end-diastolic diameter; LVESD: left ventricular end-systolic diameter; LVEDV: left ventricular end-diastolic volume; LVESV: left ventricular end-systolic volume; RVEDd: right ventricular end-diastolic diameter; TSH: thyroid-stimulating hormone; AF: atrial fibrillation. sST2.0: soluble ST2 at baseline. sST2 values expressed in pg/mL. P1: *p* value for the differences between ECV and PVI groups. P2: *p* value for the differences between AF patients and control. * *p* < 0.05.

**Table 2 ijms-24-14045-t002:** AF recurrence according to the procedure.

		Recurrence from 0 to 3 Months FU		Recurrence from 3 to 6 Months FU		Recurrence from 0 to 6 Months FU	
		NO n = 172	Yes n = 78	NO n = 186	Yes n = 64	NO n = 146	Yes n = 104
Paroxysmal	ECV	-	-	-	-	-	-
	PVI	60	8	60	8	54	14
Persistent	ECV	47	47	53	41	36	58
	PVI	65	23	73	15	56	32

ECV: electrical cardioversion. PVI: pulmonary vein isolation. FU: follow-up.

**Table 3 ijms-24-14045-t003:** sST2 values and AF recurrences in ECV patients.

		**sST2.0**	**sST2.1**	** *p* **
3-month FU	No recurrence FU(n = 47)	18,598 ± 10,916	14,680 ± 7561	0.002 *
	Recurrence (n = 47)	15,729 ± 6768	16,133 ± 7218	0.618
		**sST2.0**	**sST2.2**	** *p* **
6-month FU	No recurrence (n = 53)	16,313 ± 10,114	13,931 ± 7211	0.019 *
	Recurrence FU2 (n = 41)	18,263 ± 7611	18,950 ± 9402	0.559
		**sST2.1**	**sST2.2**	** *p* **
3- vs. 6-month FU	No recurrence (n = 53)	13,350 ± 5941	13,931 ± 7211	0.286
	Recurrence (n = 41)	18,066 ± 8256	18,950 ± 9402	0.277

Values expressed in pg/mL, as mean ± standard deviation. sST2.0: sST2 at basal, sST2.1: sST2 at 3-month FU. sST2.2: sST2 at 6-month FU. AF: atrial fibrillation, ECV: electrical cardioversion. * *p* < 0.05.

**Table 4 ijms-24-14045-t004:** sST2 values and AF recurrences in PVI patients.

		**sST2.0**	**sST2.1**	** *p* **
3-month FU	No recurrence (n = 125)	13,033 ± 7408	14,109 ± 8108	0.05
	Recurrence (n = 31)	13,765 ± 6501	14,263 ± 6430	0.565
		**sST2.0**	**sST2.2**	** *p* **
6-month FU	No recurrence (n = 133)	13,302 ± 7431	14,302 ± 6945	0.022 *
	Recurrence (n = 23)	12,461 ± 5837	13,571 ± 4181	0.451
		**sST2.1**	**sST2.2**	** *p* **
3- vs. 6-month FU	No recurrence (n = 133)	14,301 ± 8130	14,302 ± 6945	0.997
	Recurrence (n = 23)	13,208 ± 5405	13,571 ± 4181	0.728

Values expressed in pg/mL, as mean ± standard deviation. sST2.0: sST2 at basal. sST2.1: sST2 at 3-month FU. sST2.2: sST2 at 6-month FU. AF: atrial fibrillation. PVI: pulmonary vein isolation. * *p* < 0.05.

**Table 5 ijms-24-14045-t005:** (A) Logistic regression model for AF recurrence at 3-month follow-up in patients undergoing ECV. (B) Logistic regression model for AF recurrence at 6-month follow-up in patients undergoing ECV.

**(A)**				
	**OR**	**CI 95%**		** *p* **
Age (years)	0.980	0.927	1.035	0.223
Gender—male	0.443	0.123	1.503	0.165
LA index volume (mL/m^2^)	1.028	1.007	1.051	0.006 *
BMI (kg/m^2^)	1.008	0.911	1.121	0.674
Rate (bpm)	1.006	0.981	1.030	0.145
sST2.0 (pg/mL)	0.999	0.999	1.000	0.357
**(B)**				
	**OR**	**CI 95%**		** *p* **
Age (years)	1.054	0.991	1.127	0.203
Gender—male	2.168	0.863	3.530	0.751
LA index volume (mL/m^2^)	0.986	1.009	1.034	0.122
BMI (kg/m^2^)	0.940	1.009	1.034	0.417
Rate (bpm)	0.997	0.963	1.033	0.809
Rhythm at 3 months: AF	7.403	2.188	28.192	0.001 *
sST2.1 (pg/mL)	1.345	1.085	1.823	0.001 *

AF: atrial fibrillation. ECV: electrical cardioversion. LA: left atrium. BMI: body mass index. sST2.0: soluble ST2 at baseline. sST2.1: soluble ST2 at 3-month follow-up. * *p* < 0.05.

**Table 6 ijms-24-14045-t006:** (A) Logistic regression model for AF recurrence at 3 months in patients undergoing PVI. (B) Logistic regression model for AF recurrence at 6 months in patients undergoing PVI.

**(A)**				
	**OR**	**CI 95%**		** *p* **
Age (years)	0.974	0.924	1.026	0.330
Gender—male	0.261	0.065	0.974	0.049 *
BMI (kg/m^2^)	0.916	0.802	1.033	0.281
Rate (bpm)	0.956	0.918	0.989	0.019 *
Rhythm at basal: AF	5.702	1.289	6.497	0.023 *
Type AF (persistent)	2.122	0.505	9.248	0.302
sST2.0	1.000	0.999	1.000	0.353
LA index volume (mL/m^2^)	1.027	1.009	1.047	0.047 *
Low-voltage area (%)	1.008	0.986	1.032	0.376
**(B)**				
	**OR**	**CI 95%**		** *p* **
Age (years)	1.017	0.962	1.081	0.582
Gender—male	1.166	0.305	4.868	0.799
BMI (kg/m^2^)	1.036	0.898	1.188	0.625
Rate (bpm)	0.969	0.915	1.023	0.171
Rhythm at 3 months: AF	3.001	2.919	4.262	0.004 *
Type of AF (persistent)	2.247	0.164	6.290	0.625
sST2.1 (pg/mL)	0.999	0.999	0.9999	0.390
LA index volume (mL/m^2^)	0.999	0.999	1.000	0.209
Low-voltage area (%)	1.018	0.989	1.047	0.288

AF: atrial fibrillation. PVI: pulmonary vein isolation. BMI: body mass index. LA: left atrium. sST2.0: soluble ST2 at baseline. * *p* < 0.05. Rhythm: sinus rhythm as reference. Type AF: paroxysmal as reference. sST2.1: soluble ST2 at 3-month follow-up.

**Table 7 ijms-24-14045-t007:** Analysis of the ST2S biomarker at baseline in relation to the LA low-voltage area.

Low-Voltage Area (%)	ST2S.0 (pg/mL)
<5 (n = 49)	11,324 ± 5698
5 to 35 (n = 62)	14,294 ± 7621
>35 (n = 17)	10,983 ± 6679

Values expressed in pg/mL, as mean ± standard deviation. LA: left atrium. ST2S: ST2 at baseline.

## Data Availability

The datasets generated and/or analyzed during the current study are available from the corresponding author on reasonable request.

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
