# Peer review of "Role of Soluble ST2 Biomarker in Predicting Recurrence of Atrial Fibrillation after Electrical Cardioversion or Pulmonary Vein Isolation"

_ijms, 2023, doi:10.3390/ijms241814045_

Round 1
Reviewer 1 Report
Dear Authors, please find attached the review report.

Needs improvement.
Reviewer 2 Report
The authors evaluated aims the predictive value of the sSTS2 biomarker
in AF recurrence.
I have the following concerns:
1. Please include pattern of AF in the baseline characteristics.
2. What was the CRP and NT-proBNP value in the studied population?
3. Were the patients with thyroid dysfunction excluded?
4. How many patients have a history of catheter ablation?
5. AUC of 0.669 is a weak discrimination what seriously limits the study.
Minor editing of English language required.
Reviewer 3 Report
The manuscript entitled Role of soluble ST2 biomarker in predicting recurrence of atrial fibrillation after electrical cardioversion or pulmonary vein isolation is a small prospective, observational study. The authors aimed to determine the predictive value of the soluble suppression of tumorigenesis-2 (sSTS2 - a member of the IL1 receptor family) biomarker in AF recurrence in patients underwent electrical cardioversion or endocardial pulmonary vein isolation. They concluded that in patients who underwent electrical cardioversion (but not in those with pulmonary vein isolation) sST2 values at 3 months are useful to predict atrial fibrillation recurrence at 6 months of follow up.
It is well known that an increased level of sST2, this biomarker of inflammatory diseases, is predictive of adverse events [all-cause death and heart failure hospitalizations] in patients with heart failure. It is interesting that the authors taking into consideration heart rate and persistent verso paroxysmal atrial fibrillation when analyzing the level of sST2. By the way, the authors stated that high heart rate and overload due to atrial myopathy linked to the time in persistent AF are the predominant factors in the basal sST2 levels. Baseline ST2S did not show predictive capacity for AF recurrence due to the substantial baseline variability of the sST2 biomarker and follow up duration. However, this is still in debate.
This article is very interesting for clinical practice. It is well written. Discussions are very well structured.
However, I have some comments.
Why the authors excluded only heart failure with left ventricular systolic dysfunction? More than half of patients with heart failure have preserved left ventricular systolic function. Could be this an important study limitation. Please argue this comment.
In table 1 glucose was significant statistically different between study and control group. How was HbA1C in these patients? I recommend adding in table 1. Prediabetes (defined by HbA1C) seems to be an important parameter in these patients. Could be influenced the sST2 levels by the renal function and the glucose metabolism according with data literature? These remarks could be another essential limitation of the study strongly related with the above remark.
In figure 1 the acronyms are not explained.
Round 2
Reviewer 1 Report
Thank you for the revision. I have no more major remarks.
Nothing to add.
Reviewer 2 Report
I thank authors for addressing my concerns. Unofortunately I still believe that AUC of 0.669 is a weak discrimination what seriously limits the study.
Minor editing of English language required